# Mediators of Lifestyle Behaviour Changes in Obese Pregnant Women. Secondary Analyses from the DALI Lifestyle Randomised Controlled Trial

**DOI:** 10.3390/nu11020311

**Published:** 2019-02-01

**Authors:** Mireille N. M. van Poppel, Judith G. M. Jelsma, David Simmons, Roland Devlieger, Goele Jans, Sander Galjaard, Rosa Corcoy, Juan M. Adelantado, Fidelma Dunne, Jürgen Harreiter, Alexandra Kautzky-Willer, Peter Damm, Elisabeth R. Mathiesen, Dorte M. Jensen, Lise-Lotte Andersen, Mette Tanvig, Annunziata Lapolla, Maria-Grazia Dalfra, Allessandra Bertolotto, Ewa Wender-Ozegowska, Agnieszka Zawiejska, David Hill, Gernot Desoye, Frank J. Snoek

**Affiliations:** 1Institute of Sport Science, University of Graz, 8010 Graz, Austria; 2Amsterdam UMC, Vrije Universiteit Amsterdam, Department of Public and Occupational Health, Amsterdam Public Health research institute, 1081 BT Amsterdam, The Netherlands; J.Jelsma@vumc.nl; 3Western Sydney University, Campbelltown, Sydney, NSW 2760, Australia; da.simmons@westernsydney.edu.au; 4KU Leuven Department of Development and Regeneration: Pregnancy, Fetus and Neonate, Gynaecology and Obstetrics, University Hospitals Leuven, 3000 Leuven, Belgium; roland.devlieger@uz.kuleuven.ac.be (R.D.); Goele.jans@ucll.be (G.J.); s.galjaard@erasmusmc.nl (S.G.); 5Department of Obstetrics and Gynaecology, Division of Obstetrics and Prenatal Medicine, Erasmus MC, University Medical Centre Rotterdam, 3015 GD Rotterdam, The Netherlands; 6CIBER Bioengineering, Biomaterials and Nanomedicine, Instituto de Salud Carlos III, 50018 Zaragoza, Spain; RCorcoy@santpau.cat; 7Institut de Recerca de l’Hospital de la Santa Creu i Sant Pau, 08025 Barcelona, Spain; JAdelantado@santpau.cat; 8Galway Diabetes Research Centre (GDRC) and National University of Ireland, H91 CF50 Galway, Ireland; fidelma.dunne@nuigalway.ie; 9Gender Medicine Unit, Endocrinology and Metabolism, Dept. Internal Medicine III, Medical University of Vienna, 1090 Vienna, Austria; juergen.harreiter@meduniwien.ac.at (J.H.); alexandra.kautzky-willer@meduniwien.ac.at (A.K.-W.); 10Center for Pregnant Women with Diabetes, Departments of Endocrinology and Obstetrics, Rigshospitalet and The Clinical Institute of Medicine, Faculty of Health and Medical Sciences, University of Copenhagen, DK-1165 Copenhagen, Denmark; peter.damm@regionh.dk (P.D.); elisabeth.mathiesen@rh.regionh.dk (E.R.M.); 11Steno Diabetes Center Odense, Odense University Hospital, 5000 Odense, Denmark; Dorte.Moeller.Jensen@rsyd.dk; 12Department of Gynecology and Obstetrics, Odense University Hospital, 5000 Odense, Denmark; lise.lotte.andersen@rsyd.dk; 13Department of Clinical Research, Faculty of Health Science, University of Southern Denmark, 5230 Odense, Denmark; mette.tanvig@rsyd.dk; 14Department of Medical and Surgical Sciences. Università degli Studi di Padova, 35100 Padua, Italy; annunziata.lapolla@unipd.it (A.L.); mariagrazia.dalfra@sanita.padova.it (M.-G.D.); 15Department of Clinical and Experimental Medicine, University of Pisa, 56126 Pisa, Italy; alessandrabertolotto1959@yahoo.it; 16Department of Reproduction, Poznan University of Medical Sciences, 61-701 Poznan, Poland; ewaoz@post.pl (E.W.-O.); agazaw@post.pl (A.Z.); 17Recherche en Santé Lawson SA, 9952 St. Gallen, Switzerland; David.Hill@LawsonResearch.Com; 18Department of Obstetrics and Gynecology, Medical University, 8036 Graz, Austria; gernot.desoye@medunigraz.at; 19Department of Medical Psychology, Amsterdam Public Health research institute, Amsterdam University Medical Centers, VU University, 1081 HZ Amsterdam, The Netherlands; FJ.Snoek@Vumc.nl

**Keywords:** lifestyle intervention, behaviour change, mediation, pregnancy, obesity, gestational diabetes

## Abstract

A better understanding of what drives behaviour change in obese pregnant overweight women is needed to improve the effectiveness of lifestyle interventions in this group at risk for gestational diabetes (GDM). Therefore, we assessed which factors mediated behaviour change in the Vitamin D and Lifestyle Intervention for GDM Prevention (DALI) Lifestyle Study. A total of 436 women, with pre-pregnancy body mass index ≥29 kg/m^2^, ≤19 + 6 weeks of gestation and without GDM, were randomised for counselling based on motivational interviewing (MI) on healthy eating and physical activity, healthy eating alone, physical activity alone, or to a usual care group. Lifestyle was measured at baseline, and at 24–28 and 35–37 weeks of gestation. Outcome expectancy, risk perception, task self-efficacy and social support were measured at those same time points and considered as possible mediators of intervention effects on lifestyle. All three interventions resulted in increased positive outcome expectancy for GDM reduction, perceived risk to the baby and increased task self-efficacy. The latter mediated intervention effects on physical activity and reduced sugared drink consumption. In conclusion, our MI intervention was successful in increasing task self-efficacy, which was related to improved health behaviours.

## 1. Introduction

The prevalence of maternal obesity in pregnancy has increased in the past decades [1,2] and is associated with increased risks for both mother and offspring. In particular, women with obesity and women with excess gestational weight gain have an increased risk for developing gestational diabetes (GDM) [3,4]. To date, lifestyle interventions aimed to reduce the risk of GDM in women with obesity have shown only limited success [5], but do have the potential to improve lifestyle behaviours [6,7,8] and to modestly limit gestational weight gain [5].

The Vitamin D and Lifestyle Intervention for GDM Prevention (DALI) Lifestyle Study was designed to offer lifestyle counselling to obese pregnant women, in order to prevent GDM. In the DALI Lifestyle Study, the women were counselled based on seven healthy eating messages and/or five physical activity messages [9]. The counsellors used principles of motivational interviewing [10] to enhance intrinsic motivation and action planning for healthy eating and/or physical activity. We previously reported positive changes in some lifestyle behaviours and a relevant reduction in gestational weight gain in a combined healthy eating and physical activity intervention [11]. Although no improvements in glucose metabolism were observed [11], the combined healthy eating and physical activity intervention was cost-effective [12], and effective in reducing neonatal adiposity [13]. Combined, these effects warrant further implementation of the DALI combined intervention, or a similar intervention, for obese pregnant women. Therefore, understanding mechanisms behind behaviour change is important for improving future interventions.

The DALI protocol and measures were informed by the Health Action Process Approach (HAPA) [14], a theory of behaviour change, widely applied in lifestyle interventions, including those aimed at prevention of type 2 diabetes [15]. Recently, it was also found to be useful in explaining physical activity levels of women with a recent history of GDM [16]. HAPA provides a framework for understanding the process of adoption, initiation and maintenance of health behaviours. In the motivation phase preceding the volition phase (where actual behaviour change is realised), high-risk perception, positive outcome expectancies and perceived self-efficacy are regarded as three key determinants. Once intention to change is formed, self-efficacy and perceived social support remain important mediators in the process of action planning and maintenance of behaviour change [14,17].

Here, we report on secondary analyses, with the aim to enhance our understanding about why obese pregnant women change, or do not change, their behaviour. This should help us further improve lifestyle interventions for this particular patient group at high risk of GDM. As determinants of behaviour change might differ between separate lifestyle behaviours, we aimed to assess which factors mediated behaviour change for each lifestyle factor, separately, in the DALI Lifestyle Study.

## 2. Materials and Methods

### 2.1. Design and Participants

The DALI Lifestyle Study was a multi-centre randomised trial conducted in nine European countries (Austria, Belgium, Denmark (Odense, Copenhagen), Ireland, Italy (Padua, Pisa), Netherlands, Poland, Spain and United Kingdom) during 2012–2015. Local ethics committee approval and written informed consent for all women was obtained. Pregnant women with a pre-pregnancy body mass index (BMI) of ≥29 kg/m^2^, ≤19 + 6 weeks of gestation, a singleton pregnancy and aged ≥18 years were invited to participate. Exclusions included diagnosis with early gestational diabetes [18], pre-existing diabetes and chronic medical conditions [9]. Measurements occurred at baseline (prior to 20 weeks of gestation), and at 24–28 and 35–37 weeks of gestation.

### 2.2. Randomisation and Interventions

Women were randomised to a healthy eating and physical activity (HE&PA), healthy eating alone (HE), physical activity alone (PA) group or a usual care (UC) group, using a computerised random number generator, pre-stratified for site. Staff involved with measurements, but not participants, were blinded to the intervention.

In the intervention groups, participants were assigned to a single coach, with whom they discussed five physical activity and/or seven healthy eating messages (See Appendix A), depending on group allocation, and were advised to keep gestational weight gain below 5 kg. The 5 kg weight gain target—without gestational weight loss—rather than the 9 kg recommended by the Institute of Medicine [19] was chosen since it was associated with less adverse outcomes in previous studies [20,21]. Coaching, guided by principles of motivational interviewing, took place during five face-to-face sessions of 30–45 min each, alternated with up to four optional telephone calls. In the UC group, participants received no DALI interventions.

### 2.3. Outcomes

Levels of physical activity were assessed using the Pregnancy Physical Activity Questionnaire (PPAQ) [22] at the three time points. The original PPAQ consisted of 32 activities including household/caregiving, occupational, sports/exercise and inactivity measured during the current trimester. Open-ended questions allowed the respondent to add activities not already listed. In DALI, two questions for cycling to work (4.0 MET (metabolic equivalent of task)) and recreational cycling (8.0 MET) were added, due to the frequent performance of these activities in some European countries. Participants were asked to select the category that best approximated the amount of time spent in one of these activities. The duration of time spent in each activity was multiplied by its intensity such that an average weekly energy expenditure (MET hours/week) was calculated for each activity. For the open-ended reported activities, a compendium of physical activities was used to obtain MET values for intensity [23]. Activities were categorised by intensity (sedentary, light, moderate, vigorous), type and total activity (sum of all activity with an intensity above 1.5 MET). Self-reported moderate and vigorous activity were summed and presented as MET hours/week of moderate-to-vigorous physical activity.

Nutrition was assessed using a bespoke short food frequency questionnaire covering key foods linked to the intervention messages and based upon prior work [24]. The number of portions/week for each key food component (sugared drink, vegetable, carbohydrate, protein, fibre, and fat consumption) was calculated as the product of the frequency consumed/week and the number of portions per episode. A portion was the amount of food that can fit into the palm of your hand or 200 mL of fluid (a medium sized cup or glass). Portion size was calculated as the number of portions of food components per week.

The outcomes reported in this paper were limited to the six lifestyle factors that changed significantly from baseline to 24–28 weeks or 35–37 weeks of gestation due to the either the HE&PA, HE or PA intervention (moderate-to-vigorous physical activity, sedentary behaviour, sugared drink consumption, vegetable consumption, carbohydrate intake and portion size).

### 2.4. Potential Mediators

Based on the HAPA model [14], perceived risk, outcome expectancy, social support and self- efficacy relating to weight control, physical activity and nutrition were assessed at all three time points. Perceived risk for developing GDM was assessed with the following question: “What do you think is your risk of developing GDM?” Answering options ranged from 1 (very low) to 5 (very high). Outcome expectancy, social support and self-efficacy were asked with multiple single statements, as have been used in multiple previous studies, with possible answering options ranging from 1 (fully disagree) to 10 (fully agree). For example, “I believe that increasing my physical activity level will reduce my risk for developing GDM” (fully disagree–fully agree). The full list of items can be found in Appendix A.

For outcome expectancy for GDM risk reduction and for outcome expectancy for a reduction of risk to the baby, the scores of two items concerning managing weight and staying active were combined. For instance, for outcome expectancy for GDM risk reduction, the score for the item “*Managing my weight during this pregnancy, will help me to reduce my risk of developing GDM”* and the score for the item *“Staying physically active during this pregnancy, will help me to reduce my risk of developing GDM*” were summed and divided by 2 (see also Appendix A). The combined variable was then tested as potential mediator for intervention effects on sedentary behaviour and moderate-to-vigorous physical activity. Similarly, the scores for the items concerning weight management and healthy eating were averaged, and the combined variable tested as mediator of intervention effects on nutrition outcomes. For task self-efficacy, the three items for the confidence in being able to control weight, to be physically active in the short and in the long term were averaged, as well as the three items for weight control and healthy nutrition in the short and the long term.

### 2.5. Statistical Analyses

Differences between intervention groups were tested using ANOVA for continuous variables, or chi-square or Fisher’s exact tests for categorical variables. When significant differences were detected with ANOVA, the intervention groups were compared to UC group with Tukey’s post-hoc tests. All analyses were performed in SPSS version 22 (IBM Corp, Armonk, NY, USA). A two-sided *p* < 0.05 was considered significant for differences between groups. No adjustments for multiple comparisons were made.

Mediation of intervention effects at 24–28 weeks and at 35–37 weeks by different psychological constructs at 24–28 and 35–37 weeks was assessed in parallel mediation analyses [25,26]. Multiple mediation models are preferred over simple mediation models as most effects operate through multiple mechanisms at once. Furthermore, this allows the determinations of the strongest indirect effect [25].

The PROCESS macro (version 2.16) was run to compute the following steps simultaneously: (1) estimation of the intervention effects on the mediators by regressing the mediator score on the baseline mediator score, confounders and intervention dummies (a-coefficient); (2) estimation of whether changes in the mediators predict changes in the outcome variables by regressing the outcome variables onto the baseline values of the mediators, mediators, baseline values of the lifestyle outcome and intervention dummies (b-coefficient); (3) estimation of the indirect effect (a × b-coefficient); (4) estimation of the direct effect when accounted for the indirect effect (c’-path); (5) estimation of the total intervention effect (c-path). The results of these intervention effects on lifestyle outcomes vary somewhat from those previously published [11], due to differences in sample size and analysis method. For all mediation analyses, clustering per site was taken into account.

Bias-corrected bootstrapped 95% asymmetric confidence intervals (based on 10,000 bootstrap samples) were computed for the indirect effect [25]. The indirect effect is statistically different from zero if the confidence interval does not straddle zero. PROCESS was run for each lifestyle outcome with the same value for the random number seed attributed to each run. This was done to obtain results as if all the paths were estimated in one model with multiple dependent variables. The intervention condition is a multi-categorical independent variable and was therefore analysed with dummy coding [26].

## 3. Results

Maternal baseline characteristics were comparable between groups (Table 1). Psychological variables and lifestyle behaviours per intervention group are described in Table 2 for all three time points in pregnancy. No baseline differences in psychological variables were found. Women’s expectancy—that managing weight, increasing physical activity and healthy eating would lead to a reduction in GDM risk and in a risk reduction for their baby—was high at the start of the study. Task self-efficacy for healthy eating was somewhat higher than for physical activity.

### 3.1. Intervention Effects on Psychological Variables (A-Path)

Intervention effects on psychological variables at 24–28 and 35–37 weeks can be found in Table 3 and Figure 1, Figure 2 and Figure 3 (a-path). Small variations in results are due to differences in sample sizes in the various analyses. In general, all three interventions increased the women’s expectancy that managing weight, increasing physical activity and healthy eating would lead to a reduction in GDM risk and in a risk reduction for their baby. Furthermore, both the HE&PA and PA intervention increased task self-efficacy for physical activity. Task self-efficacy for healthy eating was increased in the HE&PA and HE group. In the HE&PA intervention, increased satisfaction with social support was observed for physical activity. Satisfaction with social support for healthy eating increased in the HE group, but was only statistically significant in some of the analyses, depending on the outcome variable tested in the mediation model and the number of participants having missing data for this specific outcome variable. None of the interventions had an effect on the perceived risk of developing GDM.

### 3.2. Association of Psychological Variables with Lifestyle Behaviours (B-Path)

Task self-efficacy for physical activity at 24–28 weeks was positively associated with moderate-to-vigorous physical activity (MVPA) at 24–28 (0.04; 95% confidence interval [CI] 0.02, 0.07) and 35–37 weeks (0.03; 95% CI 0.02, 0.06) (Figure 1A,B; Appendix A). Outcome expectancy of a reduction in GDM risk at 35–37 weeks was positively associated with MVPA at the same time point (0.04; 95% CI 0.001, 0.07, *p* = 0.047). Task self-efficacy for healthy eating at 24–28 and 35–37 weeks was negatively associated with sugared drink consumption at the same time point (for 24–28 weeks −0.24; 95% CI −0.42, −0.06; for 35–37 weeks −0.25; 95% CI −0.41, −0.09). The association of task self-efficacy at 24–28 weeks with sugared drink consumption at 35–37 weeks was not significant (−0.24; 95% CI −0.51, 0.04) (Appendix A). Increased satisfaction with social support for healthy eating at 24–28 weeks was negatively associated with portion size at 35–37 weeks (−1.15; 95% CI −2.30, −0.01) (Appendix A). Outcome expectancy of reduced risk for their baby at 35–37 weeks (0.79; 95% CI 0.09, 1.48) and task self-efficacy at 35–37 weeks (−0.55; 95% CI −0.95, −0.15) were associated with portion size at 35–37 weeks. No other significant associations between psychological variables and lifestyle behaviours were observed (Appendix A).

### 3.3. Mediation by Psychological Variables of Intervention Effects on Physical Activity and Sedentary Behaviour (a × b path)

Appendix A shows the total intervention effects on moderate-to-vigorous physical activity (MVPA) and sedentary behaviour (c-path) at 24–28 and at 35–37 weeks. Task self-efficacy for physical activity at 24–28 weeks was a mediator of intervention effects on MVPA at 24–28 weeks in both the HE&PA (indirect effect 0.09 (95% CI 0.03; 0.19)) and the PA group (indirect effect 0.08 (95% CI 0.02; 0.18)). Also, at 35–37 weeks, task self-efficacy at 24–28 weeks mediated intervention effects on MVPA in those two groups (HE&PA: indirect effect 0.06 (95% CI 0.01; 0.16); PA: indirect effect 0.08 (95% CI 0.01; 0.20)) (Figure 1A,B, Appendix A). Outcome expectancy of reduction of GDM risk at 35–37 weeks mediated intervention effects on MVPA at 35–37 weeks in the PA group (indirect effect 0.07 (95% CI 0.001; 0.22). None of the psychosocial variables at 24–28 or 35–37 weeks mediated intervention effects on sedentary behaviour at 24–28 or at 35–37 weeks (Appendix A).

### 3.4. Mediation by Psychological Variables of Intervention Effects on Healthy Eating Behaviours (a × b path)

Appendix A shows the total intervention effects on sugared drink consumption, vegetable consumption or carbohydrate intake, and portion size (c-path) at 24–28 and at 35–37 weeks. For sugared drink consumption at 24–28 weeks, two mediating variables were found with opposite effects in the HE&PA and HE intervention groups (Figure 2, Appendix A). The expectancy that weight management and healthy eating would result in a risk reduction for the baby was increased in all three intervention groups. This outcome expectancy was however associated with *increased* sugared drink consumption, resulting in significant indirect effects in all three groups (indirect effects HE&PA: 0.38 (95% CI 0.02, 1.13), HE: 0.38 (95% CI 0.01, 1.21), PA: 0.43 (95% CI 0.02, 1.32)). At the same time, task self-efficacy for healthy eating increased in the HE&PA and HE groups. Higher task self-efficacy was associated with *reduced* sugared drink consumption, resulting in significant indirect effects in the HE&PA (−0.56 (95% CI −1.43, −0.07)) and HE group (−0.48 (95% CI −1.27, −0.06)) (Figure 2, Appendix A). Higher task self-efficacy at 35–37 weeks also mediated intervention effects on sugared drink consumption at 35–37 weeks in the HE&PA group (indirect effect −0.46 (95% CI −1.19; −0.05)).

Outcome expectancy of a risk reduction for the baby at 24–28 was a mediating variable for the intervention effect on portion size at 24–28 weeks in all three intervention groups (Figure 3A, Appendix A), with indirect effects varying from 0.80 (95% CI 0.05, 2.47) in HE&PA to 0.97 (95% CI 0.07, 2.73) in the HE group. At 35–37 weeks, satisfaction with social support for healthy eating at 24–28 weeks mediated intervention effects in the HE group (indirect effect −0.61 (95% CI −1.93, −0.03) for portion size (Figure 3B, Appendix A). Furthermore, outcome expectancy of a risk reduction for their baby and task self-efficacy at 35–37 weeks mediated intervention effects on portion size at 35–37 weeks, both in the HE&PA and HE groups (for outcome expectancy HE&PA: 0.96 (95% CI 0.06; 2.73), HE: 0.64 (95% CI 0.02, 2.10) and for task self-efficacy HE&PA: −1.03 (95% CI −2.72, −0.12), HE: −0.85 (95% CI −2.24, −0.11)).

None of the psychological variables at 24–28 or 35–37 weeks mediated intervention effects on vegetable consumption (Appendix A) or carbohydrate intake (Appendix A) at 24–28 or at 35–37 weeks.

## 4. Discussion

We found that all three intervention groups in the DALI Lifestyle Study were associated with increased outcome expectancy for the reduction of GDM risk and the risk to the baby, and with increased task self-efficacy compared to the usual care group. Furthermore, although the interventions were not specifically aimed at this, women were more satisfied with the social support they received for being physically active in the combined HE&PA group and for healthy eating in the HE group. The improvements in task self-efficacy were related to increased physical activity, reduced sugared drink consumption and portion size. Increased satisfaction with social support for healthy eating was related to decreased portion size. Unexpectedly, higher outcome expectancy of a risk reduction for the baby was related to increased sugared drink consumption and portion size.

Our findings confirm the importance of self-efficacy for behaviour change. Self-efficacy is well established as determinant of a healthy lifestyle and lifestyle changes. Recently, the importance of self-efficacy as predictor of physical activity was shown in women with a history of GDM [16]. We now demonstrate the importance of self-efficacy both for moderate-to-vigorous physical activity and for sugared drink consumption for obese pregnant women at risk of GDM. However, a review of mechanisms of change specifically within motivational interviewing interventions found no evidence for a mediating effect of self-efficacy in relation to behaviour change [27]. The authors found this surprising since “self-efficacy is an important construct in motivational interviewing” and suggested this lack of evidence might have been due to poor measurement of self-efficacy in previous studies. We made sure that we linked the questions on self-efficacy for reaching weight, physical activity and dietary goals to the relevant behaviours. Furthermore, we measured it on a 10-point scale, and did not have problems with ceiling effects in our population. This might explain why we found significant mediation of self-efficacy of effects on moderate-to-vigorous physical activity and sugared drink consumption after a motivational interviewing intervention. However, the statements for measuring self-efficacy and, also, the other psychological variables, might have lacked specificity for the other behaviours (sedentary behaviour, vegetable and carbohydrate intake) for which no mediation by self-efficacy was found. For future interventions, the behaviour change techniques most suited for improving task self-efficacy should be identified and integrated in the intervention, as suggested by Ainscough et al., who developed a decision tree to support the development of antenatal lifestyle interventions [28].

Although the interventions increased the outcome expectancy that managing weight and obtaining physical activity and nutrition goals would lead to a reduction in the risk for GDM, this was not found to mediate intervention effects on lifestyle outcomes at 24–28 weeks, and only the effect on moderate-to-vigorous physical activity at 35–37 weeks. This is partly in line with previous findings in women with a history of GDM [16] and patients with diabetes [29], in whom outcome expectancy was not associated with the intention to be physically active. More research is warranted to elucidate the role of outcome expectancies in prevention trials. More so, as we did find mediation by outcome expectancy of a reduction of risk to the baby for intervention effects on sugared drink consumption and portion size, but in an unexpected direction: increased outcome expectancy led to less beneficial behaviour in both cases. We have no plausible explanation for this finding. Future qualitative research may help to understand the link between outcome expectancies with regard to the baby’s health and changes in eating behaviour.

In the DALI Lifestyle Study, risk perception of developing GDM was not changed by the interventions. Overall, risk perception decreased somewhat over time, possibly after the majority of women tested negatively for GDM at baseline and 24–28 weeks of gestation. In line with previous studies, risk perception was not a predictor of behaviour change [16,30,31], which questions the usefulness of this construct in the context of preventing GDM. According to the HAPA model, risk perception is important for intention formation (i.e., in the motivation phase), and not in the volition phase. Therefore, it is likely that in the context of our DALI Lifestyle Study, in which women participated voluntarily and were likely motivated for behaviour change to start with, risk perception does not play a role in behaviour change.

Although no significant intervention effects were found on sugared drink consumption at 24–28 weeks, mediation by both outcome expectancy of risk reduction for the baby and task self-efficacy was found. The mediating effects were in opposite directions (i.e., high task self-efficacy reduced sugared drink consumption and a high outcome expectancy increased consumption), which might have cancelled each other out and, therefore, limited intervention effects at this point in time. Also, for portion size mediation, satisfaction with social support without significant intervention effects in the HE&PA and PA groups was found. Apparently, there are some unmeasured mediators that work in opposite directions.

We only found limited intervention effects on satisfaction with social support for either physical activity or healthy eating, which might explain why no mediation by this factor was found. In the DALI interventions, social support was only discussed when women indicated to perceive too little support as a barrier for behaviour change. However, our findings could suggest that more attention should be paid to this aspect in the intervention, in order to achieve improvements, even when women do not raise the issue themselves.

### Strengths and Limitations

The study was grounded in behaviour change theory with key psychological constructs measured over time. While the statements we used to question participants were almost similar to the ones used in previous HAPA research, we did not test their psychometrics. However internal validity could be inferred from the fact that self-efficacy for PA increased in both the HE&PA and PA group, but not the HE group and, vice versa, self-efficacy for healthy eating increased in the HE&PA and HE groups, but not the PA group. Another limitation of our study is that lifestyle outcomes were self-reported. However, we used lifestyle outcomes that were closely related to the intervention goals, and differentiated between psychological factors related to physical activity or to healthy eating. For practical reasons, we unfortunately did not include any items for psychological factors specifically related to sedentary behaviour, or to the specific healthy eating goals. This might explain why no mediation was found for sedentary behaviour, or for vegetable and carbohydrate intake.

## 5. Conclusions

In conclusion, our motivational interviewing intervention was successful in increasing outcome expectancy and task self-efficacy, both of which were related to improvements in some dietary and physical activity behaviours. Our findings corroborate the importance of self-efficacy in the process of improving lifestyle to improve clinical outcomes. The role of outcome expectancy for behaviour change needs further study.

## Figures and Tables

**Figure 1 nutrients-11-00311-f001:**
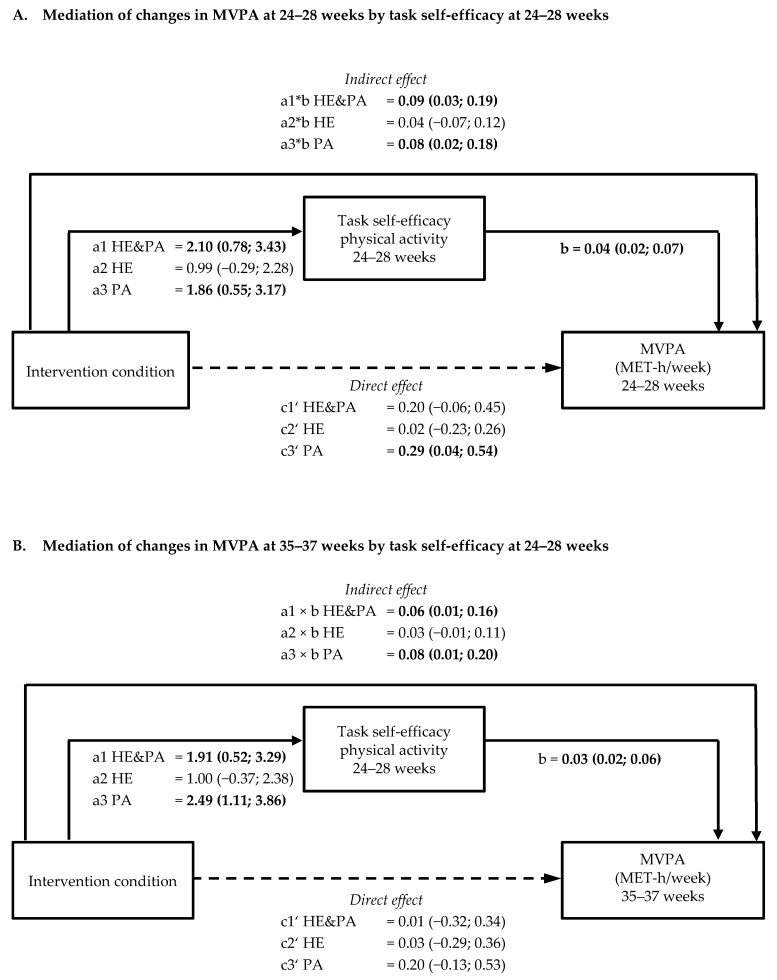
Schematic diagram of the results of mediation by task self-efficacy on moderate-to-vigorous physical activity. Shown are the path coefficients between intervention condition, task self-efficacy at 24–28 weeks, and moderate to vigorous physical activity (MVPA) at 24–28 weeks (**A**) and at 35–37 weeks (**B**). Statistically significant path coefficients are shown in bold and with solid lines. Dashed lines indicated non-significant paths. Indirect effects were calculated as the product of the coefficients of the a and b paths (a × b).

**Figure 2 nutrients-11-00311-f002:**
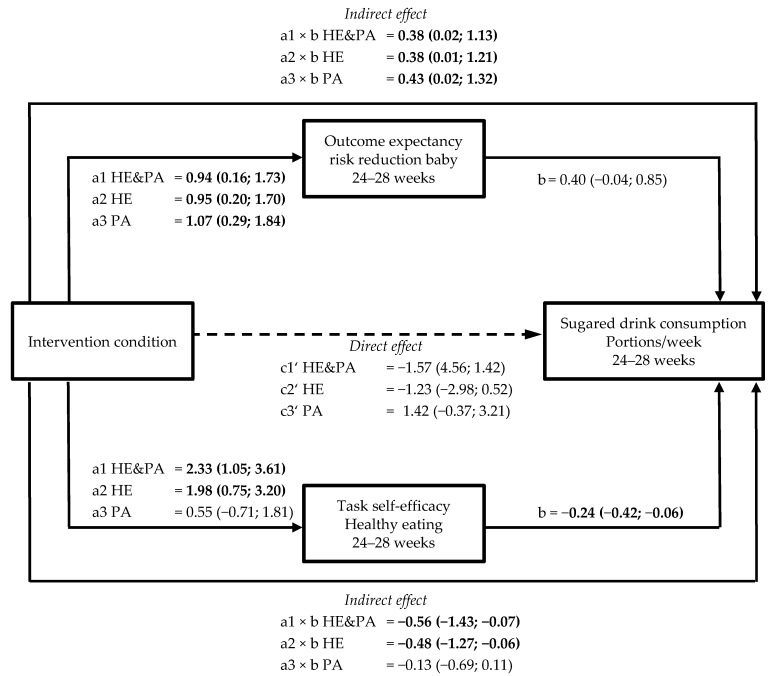
Schematic diagram of the results of mediation by outcome expectancy and task self-efficacy on sugared drink consumption. Shown are the path coefficients between intervention condition, outcome expectancy of risk reduction for the baby, and task self-efficacy at 24–28 weeks, and sugared drink consumption at 24–28 weeks. Statistically significant path coefficients are shown in bold and with solid lines. Dashed lines indicate non-significant paths. Indirect effects were calculated as the product of the coefficients of the a and b paths (a × b).

**Figure 3 nutrients-11-00311-f003:**
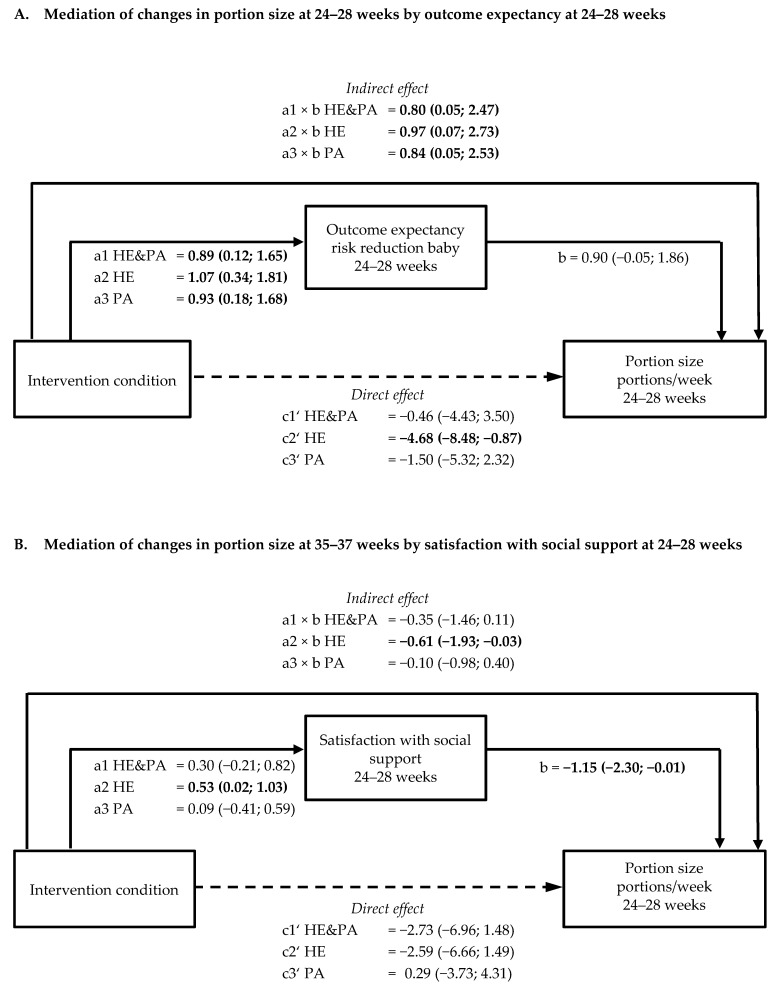
Schematic diagram of the results of mediation by outcome expectancy or satisfaction with social support on portion size. Shown are the path coefficients between intervention condition, outcome expectancy of risk reduction of the baby at 24–28 weeks, and portion size at 24–28 weeks (**A**) and between intervention condition, satisfaction with social support, and portion size at 35–37 weeks (**B**). Statistically significant path coefficients are shown in bold and with solid lines. Dashed lines indicate non-significant paths. Indirect effects were calculated as the product of the coefficients of the a and b paths (a × b).

**Table 1 nutrients-11-00311-t001:** Baseline characteristics of all women included per intervention group.

Variable	UC *n* = 105	HE&PA *n* = 108	HE *n* = 113	PA *n* = 110	Total *n* = 436	*p*
Age, year, mean ± SD	31.8 ± 5.6	31.9 ± 5.3	32.5 ± 5.5	31.7 ± 5.1	32.0 ± 5.4	0.68
Multiparous, *n* (%)	50 (48%)	56 (52%)	64 (57%)	51 (46%)	221 (51%)	0.41
European descent, *n* (%)	94 (90%)	95 (88%)	95 (84%)	94 (86%)	378 (87%)	0.65
Lives with partner, *n* (%)	100 (95%)	99 (92%)	108 (96%)	103 (94%)	410 (94%)	0.60
Higher education, *n* (%)	55 (52%)	59 (55%)	65 (58%)	60 (55%)	239 (55%)	0.90
Maternal smoking, *n* (%)	18 (17%)	11 (10%)	20 (18%)	18 (17%)	67 (15%)	0.38
History of GDM, *n* (%)	3 (5%)	4 (6%)	6 (7%)	4 (6%)	17 (6%)	0.95
1st degree FH DM, *n* (%)	28 (27%)	18 (17%)	28 (25%)	27 (25%)	101 (23%)	0.31
Chronic hypertension, *n* (%)	9 (9%)	12 (11%)	17 (15%)	17 (16%)	55 (13%)	0.36
Gestation on entry, weeks, mean ± SD	15.2 ± 2.4	15.2 ± 2.2	15.3 ± 2.5	15.5 ± 2.3	15.3 ± 2.3	0.58
Weight at entry, kg, mean ± SD	94.2 ± 12.6	95.2 ± 13.8	94.9 ± 13.2	94.6 ± 12.8	94.7 ± 13.1	0.89
Height, cm, mean ± SD	165.9 ± 6.7	166.0 ± 6.6	165.2 ± 6.6	165.6 ± 7.2	165.7 ± 6.8	0.81
BMI at entry, kg/m^2^, mean ± SD	34.2 ± 3.9	34.5 ± 4.0	34.7 ± 4.2	34.4 ± 3.8	34.5 ± 4.0	0.83

Abbreviations: UC = usual care; HE = healthy eating; PA = physical activity; HE&PA = healthy eating and physical activity; GDM = gestational diabetes; FH DM = family history of diabetes; SD = standard deviation; BMI = body mass index. *p* values of the tests for differences between groups are based on chi-square test (dichotomous variables) or ANOVA (continuous variables).

**Table 2 nutrients-11-00311-t002:** Potential mediators and lifestyle outcomes per intervention group at all three time points.

Psychological Variables	UC		HE&PA		HE		PA		
Baseline <20 Weeks	*n*	Mean ± SD	*n*	Mean ± SD	*n*	Mean ± SD	*n*	Mean ± SD	*p*
Attitude towards weight (range 2–20)	103	15.7 ± 3.4	108	15.2 ± 4.1	112	15.3 ± 3.5	107	15.9 ± 3.6	0.41
Perceived risk for GDM (range 1–5)	105	2.8 ± 0.8	107	2.8 ± 0.9	113	2.9 ± 0.8	110	2.9 ± 0.9	0.75
Outcome expectancy GDM risk reduction (range 2–20)									
Healthy eating	104	16.5 ± 3.1	106	16.6 ± 3.0	112	16.8 ± 2.9	108	17.0 ± 3.0	0.60
Physical activity	103	16.3 ± 3.0	106	16.3 ± 3.2	112	16.5 ± 3.0	106	16.8 ± 3.2	0.64
Outcome expectancy risk reduction baby (range 2–20)									
Healthy eating	104	17.3 ± 2.5	106	16.8 ± 3.0	112	17.5 ± 2.7	109	17.7 ± 2.6	0.11
Physical activity	104	16.8 ± 2.7	107	16.6 ± 3.2	112	17.3 ± 2.8	110	17.4 ± 2.8	0.09
Task self-efficacy (range 3–30)									
Healthy eating	104	21.2 ± 5.2	108	22.0 ± 5.1	112	21.3 ± 5.2	110	21.8 ± 5.0	0.63
Physical activity	104	20.7 ± 5.2	107	20.8 ± 5.1	112	20.7 ± 5.1	109	21.4 ± 5.5	0.67
Satisfaction with social support (range 1–10)									
Healthy eating	104	7.6 ± 2.0	108	7.6 ± 1.9	112	7.4 ± 2.1	109	7.8 ± 2.2	0.61
Physical activity	104	7.7 ± 2.0	108	7.5 ± 2.1	112	7.4 ± 2.3	108	7.8 ± 2.2	0.39
**24** **–** **28 weeks**									
Attitude towards weight (range 2–20)	99	13.9 ± 3.7	95	14.8 ± 3.5	104	14.7 ± 3.8	101	14.7 ± 3.5	0.30
Perceived risk for GDM (range 1–5)	100	2.6 ± 0.9	95	2.7 ± 1.0	104	2.7 ± 0.9	100	2.6 ± 1.0	0.87
Outcome expectancy GDM risk reduction (range 2–20)									
Healthy eating	100	15.5 ± 3.3	95	16.5 ± 3.6	105	**16.9 ± 3.1**	101	16.6 ± 3.2	0.01
Physical activity	100	15.1 ± 3.5	94	16.3 ± 3.7	105	16.5 ± 3.3	101	16.6 ± 3.3	0.005
Outcome expectancy risk reduction baby (range 2–20)									
Healthy eating	101	16.2 ± 3.3	94	17.0 ± 3.4	104	**17.5 ± 3.0**	101	**17.5 ± 2.5**	0.01
Physical activity	101	16.0 ± 3.5	95	16.8 ± 3.5	104	**17.2 ± 3.0**	101	**17.4 ± 2.6**	0.01
Task self-efficacy (range 3–30)									
Healthy eating	101	19.4 ± 4.7	96	**21.8 ± 4.5**	104	**21.5 ± 4.7**	100	20.5 ± 5.3	0.002
Physical activity	100	18.6 ± 5.0	96	**20.6 ± 5.0**	105	19.6 ± 5.3	101	**20.9 ± 5.6**	0.01
Satisfaction with social support (range 1–10)									
Healthy eating	101	7.5 ± 1.8	96	7.8 ± 2.0	105	7.8 ± 2.0	101	7.8 ± 2.0	0.58
Physical activity	100	7.2 ± 2.1	96	7.8 ± 2.0	105	7.6 ± 2.3	101	**8.0 ± 2.2**	0.047
**35** **–** **37 weeks**									
Attitude towards weight (range 2–20)	89	13.0 ± 4.3	86	14.1 ± 4.0	88	14.0 ± 4.1	89	14.4 ±3.9	0.14
Perceived risk for GDM (range 1–5)	86	2.3 ± 1.1	86	2.1 ± 1.2	85	2.3 ± 1.1	87	2.1 ± 1.2	0.52
Outcome expectancy GDM risk reduction (range 2–20)									
Healthy eating	88	14.2 ± 4.6	85	15.5 ± 5.3	88	15.3 ± 5.0	88	15.7 ± 4.5	0.20
Physical activity	88	13.9 ± 4.7	85	15.3 ± 5.3	87	14.9 ± 5.1	88	15.8 ± 4.6	0.09
Outcome expectancy risk reduction baby (range 2–20)									
Healthy eating	89	15.5 ± 3.6	85	**16.8 ± 3.5**	87	16.7 ± 3.0	89	16.7 ± 3.0	0.03
Physical activity	89	15.2 ± 3.9	85	16.5 ± 3.6	87	16.4 ± 3.1	89	**16.7 ± 3.1**	0.02
Task self-efficacy (range 3–30)									
Healthy eating	88	19.9 ± 4.9	83	**22.1 ± 5.5**	88	21.5 ± 4.4	89	20.8 ± 5.8	0.03
Physical activity	88	17.4 ± 5.7	84	19.4 ± 5.7	87	19.4 ± 5.7	87	**19.7 ± 6.0**	0.03
Satisfaction with social support (range 1–10)									
Healthy eating	89	7.4 ± 2.1	86	7.7 ± 2.1	88	**8.0 ± 1.7**	89	7.7 ± 2.1	0.26
Physical activity	89	7.4 ± 2.2	86	7.7 ± 2.2	86	7.9 ± 1.9	89	8.2 ± 1.9	0.10
**Lifestyle behaviours** **24** **–** **28 weeks**									
MVPA, MET-h/week *	101	29.7 (15.3; 70.3)	96	43.1 (21.0; 77.8)	104	35.7 (17.1; 83.5)	101	**51.2 (25.3; 83.5)**	0,03
Sedentary behaviour, MET-h/week	101	13.4 ± 9.7	96	10.8 ± 8.9	104	11.8 ± 7.8	101	12.3 ± 7.3	0.19
Sugary drinks, portions/week	96	4.8 ± 5.0	90	5.3 ± 7.6	99	3.2 ± 4.2	96	**7.4 ± 9.6**	<0.001
Vegetables, portions/week	97	11.6 ± 9.2	91	13.8 ± 8.5	98	**15.2 ± 9.7**	95	12.7 ± 9.2	0.04
Carbohydrates, portions /week	92	35.8 ± 19.0	87	37.0 ± 17.5	98	32.6 ± 15.6	90	35.4 ± 19.6	0.40
Portion size, portions/week	97	18.1 ± 13.1	91	19.9 ± 17.7	97	14.1 ±10.1	94	20.1 ± 14.4	0.01
**35** **–** **37 weeks**									
MVPA, MET-h/week *	89	21.6 (11.6; 42.0)	86	27.2 (7.8; 54.2)	88	21.1 (9.9; 59.6)	88	**35.4 (15.8; 60.4)**	0.18
Sedentary behaviour, MET-h/week	89	13.8 ± 9.9	86	10.7 ± 7.7	88	13.2 ± 9.6	88	13.9 ± 8.2	0.06
Sugary drinks, portions/week	77	5.4 ± 5.9	79	3.6 ± 4.7	82	3.3 ± 5.1	76	6.2 ± 6.7	0.002
Vegetables, portions/week	86	12.9 ± 9.5	81	12.6 ± 9.2	82	14.2 ± 10.1	82	12.6 ± 11.0	0.69
Carbohydrates, portions/week	77	37.3 ± 19.7	79	32.9 ± 18.4	80	**29.6 ± 13.9**	77	36.5 ± 20.4	0.03
Portion size, portions /week	81	17.9 ± 13.8	80	16.7 ± 14.1	81	14.8 ± 11.3	80	21.3 ± 17.4	0.03

* Data presented as median (interquartile range) because of skewed distribution. Abbreviations: UC = usual care; HE = healthy eating; PA = physical activity; HE&PA = healthy eating & physical activity; GDM = gestational diabetes; SD = standard deviation; MVPA = moderate-to-vigorous physical activity; MET = metabolic equivalent of task. *p* values of the tests for differences between groups are based on ANOVA (with natural log values for MVPA). Significant differences between intervention group and UC group based on Tukey’s post hoc tests are in bold.

**Table 3 nutrients-11-00311-t003:** Intervention effects on psychological variables at 24–28 and 35–37 weeks (a path).*

**24–28 Weeks**	**HE&PA vs. UC** **Estimate (95% CI)**	***p***	**HE vs. UC** **Estimate (95% CI)**	***p***	**PA vs. UC** **Estimate (95% CI)**	***p***
Perceived risk for GDM (range 1–5)	0.04 (−0.19; 0.27)	0.74	0.002 (−0.22; 0.23)	0.99	−0.01 (−0.24; 0.22)	0.92
Outcome expectancy GDM risk reduction with physical activity (range 2–20)	**1.31 (0.47; 2.14)**	0.002	**1.24 (0.43; 2.05)**	0.003	**1.38 (0.56; 2.21)**	0.001
Outcome expectancy GDM risk reduction with healthy eating (range 2–20)	**1.25 (0.43; 2.07)**	0.003	**1.24 (0.43; 2.03)**	0.003	**1.12 (0.30; 1.93)**	0.007
Outcome expectancy risk reduction baby with physical activity (range 2–20)	**0.83 (0.03; 1.63)**	0.04	**0.92 (0.14; 1.70)**	0.02	**0.96 (0.17; 1.75)**	0.02
Outcome expectancy risk reduction baby with healthy eating (range 2–20)	**0.89 (0.13; 1.66)**	0.02	**0.91 (0.16; 1.66)**	0.02	**0.95 (0.19; 1.72)**	0.01
Task self-efficacy for physical activity (range 3–30)	**2.11 (0.81; 3.42)**	0.002	1.09 (−0.18; 2.37)	0.09	**1.89 (0.59; 3.18)**	0.004
Task self-efficacy for healthy eating (range 3–30)	**2.19 (0.94; 3.43)**	0.001	**2.32 (1.10; 3.54)**	0.0002	0.49 (−0.75; 1.73)	0.44
Satisfaction with social support for physical activity (range 1–10)	**0.57 (0.06; 1.08)**	0.03	0.41 (−0.09; 0.90)	0.11	0.49 (−0.01; 1.00)	0.054
Satisfaction with social support for healthy eating (range 1–10)	0.10 (−0.38; 0.59)	0.67	0.38 (−0.09; 0.84) **	0.12	0.07 (−0.41; 0.54)	0.78
**35–37 weeks**	**HE&PA vs. UC** **Estimate (95% CI)**	***p***	**HE vs. UC** **Estimate (95% CI)**	***p***	**PA vs. UC** **Estimate (95% CI)**	***p***
Perceived risk for GDM (range 1–5)	−0.02 (−0.35; 0.31)	0.74	−0.001 (−0.33; 0.32)	0.99	−0.15 (−0.47; 0.18)	0.37
Outcome expectancy GDM risk reduction with physical activity (range 2–20)	**1.99 (0.62; 3.36)**	0.002	0.87 (−0.49; 2.22)	0.21	**1.74 (0.39; 3.10)**	0.01
Outcome expectancy GDM risk reduction with healthy eating (range 2–20)	**2.09 (0.69; 3.49)**	0.004	**0.97 (−0.41; 2.35)**	0.17	1.32 (−0.05; 2.69)	0.06
Outcome expectancy risk reduction baby with physical activity (range 2–20)	**1.29 (0.39; 2.18)**	0.04	0.85 (−0.03; 1.74)	0.06	**1.25 (0.36; 2.13)**	0.01
Outcome expectancy risk reduction baby with healthy eating (range 2–20)	**1.46 (0.56; 2.36)**	0.002	**0.82 (−0.07; 1.71)**	0.07	0.82 (−0.06; 1.71)	0.07
Task self-efficacy for physical activity (range 3–30)	**2.02 (0.38; 3.67)**	0.002	**1.98 (0.35; 3.60)**	0.02	**1.90 (0.27; 3.53)**	0.02
Task self-efficacy for healthy eating (range 3–30)	**2.01 (0.49; 3.53)**	0.01	1.43 (−0.07; 2.93)	0.06	0.37 (−1.12; 1.86)	0.63
Satisfaction with social support for physical activity (range 1–10)	**0.63 (0.07; 1.19)**	0.03	**0.63 (0.08; 1.18)**	0.03	**0.78 (0.23; 1.33)**	0.01
Satisfaction with social support for healthy eating (range 1–10)	0.54 (−0.02; 1.10)	0.06	**0.65 (0.10; 1.21)**	0.02	0.36 (−0.19; 0.91)	0.20

* Values vary slightly between analyses for different lifestyle outcomes, due to varying sample sizes. ** Satisfaction with social support for HE was statistically significant in the HE group, depending on the sample. Abbreviations: UC = usual care; HE = healthy eating; PA = physical activity; HE&PA = healthy eating and physical activity; GDM = gestational diabetes. *p* values based on regressions from the mediation analyses. Significant differences compared to the UC group are in bold.

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
