# Peer review of "Mediators of Lifestyle Behaviour Changes in Obese Pregnant Women. Secondary Analyses from the DALI Lifestyle Randomised Controlled Trial"

_nutrients, 2019, doi:10.3390/nu11020311_

Round 1
Reviewer 1 Report
Comments to authors
This manuscript addresses the issue of lifestyle behavior change and its determinants among obese pregnant women in a large intervention study. The results are somewhat limited in the identification of mediating factors of behavior change, with only improved task self-efficacy standing out as a key factor.
1. Abstract line 53: this should read “pregnant obese women” since this was the population under study in this paper.
2. Introduction line 74-76: If there is little evidence for success of lifestyle interventions in reducing GDM in obese women, what is the significance of improvements in lifestyle behaviors or reduced GWG? Need to expand on this point to justify why this research and examining mediators of behavior change is even valid. E.g. is there evidence that sustained behavior change into the postpartum period reduces intrapartum weight retention or other longer term maternal health outcomes?
3. Introduction line 97: remove the word “have” in the phrase “we have aimed…”
4. Methods line 138: It is not clear what the nutrition variable “portion size” actually represents in the analysis. It is listed alongside food group intakes which are correctly measured in portions/week, but it does not make sense that portion size can also be measured in this way. Should this measure not be grams/day? This should also be corrected in Table 2.
5. Methods line 171-172: The psychological constructs considered as potential mediators in this study were assessed at 3 time points, but only the 24-28 week assessment data are included in the mediation analysis (with consideration for baseline values in the model). But the values at the late assessment could also mediate the outcomes in late pregnancy. The authors should justify their rationale for only using the mid pregnancy values, or consider repeating the analysis using change in psychological constructs from baseline to late pregnancy.
6. Table 1: Please include p-values from ANOVA analysis to report any potential differences in characteristics between the groups which could potentially influence outcomes and psychological mediators. For example, the rate of smokers appears lower in the HE&PA group which could have a preferential influence on dietary intakes and compliance to PA goals.
7. Table 2: Again, reporting any potential differences in each construct across the groups at each time point would be helpful here. If there are any differences at baseline, this could influence the effectiveness of interventions. Any differences between groups at follow-up assessments could indicate effectiveness of a particular intervention.
8. Table 2 and 3: the range values for psychological constructs in parentheses should be explained. Either in the footnotes or by adding the term “range” before the numbers.
9. Results section 3.2: “Social support for healthy eating was negatively associated with portion size” should include “at 35-37 weeks” since the association was only significant at this time point.
10. Results section end of page 4: the caption for Figure 2 appears to be incorrectly inserted here.
11. All figures: please improve layout so that all labels and text around the arrows is visible and legible.
12. Discussion: First sentence does not make sense. It seems you mean to say the following “we found that all three interventions in the DALI study were associated with an increase in outcome expectancy… compared to the control group”.
13. Discussion: Improved self-efficacy is the dominant finding in this study in relation to mediating effects on healthy behaviors. The authors should make suggestions on how future studies that focus on lifestyle behavior change in pregnancy, or clinical practice with overweight/obese pregnant women, can incorporate efforts to increase women’s self-efficacy for more effective health outcomes. Also please reference this relevant paper in relation to how intervention studies with pregnant women can be designed to incorporate effective behavior change modalities: Ainscough et al., Public Health Nutr. 2017 Oct;20(14):2642-2648. doi: 10.1017/S136898001700129X.
14. Conclusion: in the first sentence, please re-write as “…of which the latter was related to improvements in some dietary and physical activity behaviors.”
Author Response
This manuscript addresses the issue of lifestyle behavior change and its determinants among obese pregnant women in a large intervention study. The results are somewhat limited in the identification of mediating factors of behavior change, with only improved task self-efficacy standing out as a key factor.
1. Abstract line 53: this should read “pregnant obese women” since this was the population under study in this paper.
We have added this in the abstract text.
2. Introduction line 74-76: If there is little evidence for success of lifestyle interventions in reducing GDM in obese women, what is the significance of improvements in lifestyle behaviors or reduced GWG? Need to expand on this point to justify why this research and examining mediators of behavior change is even valid. E.g. is there evidence that sustained behavior change into the postpartum period reduces intrapartum weight retention or other longer term maternal health outcomes?
We have added some information about the cost-effectiveness of the combined intervention, and about the effect on neonatal adiposity, to justify further research on this intervention.
3. Introduction line 97: remove the word “have” in the phrase “we have aimed…”
Changed as suggested.
4. Methods line 138: It is not clear what the nutrition variable “portion size” actually represents in the analysis. It is listed alongside food group intakes which are correctly measured in portions/week, but it does not make sense that portion size can also be measured in this way. Should this measure not be grams/day? This should also be corrected in Table 2.
We have changed the description of portion size in the methods section. Since we did not have information about grams or ml of each food component, we added the number of portions per week of the different food components. Therefore, the unit is indeed portions/week instead of grams/day.
5. Methods line 171-172: The psychological constructs considered as potential mediators in this study were assessed at 3 time points, but only the 24-28 week assessment data are included in the mediation analysis (with consideration for baseline values in the model). But the values at the late assessment could also mediate the outcomes in late pregnancy. The authors should justify their rationale for only using the mid pregnancy values, or consider repeating the analysis using change in psychological constructs from baseline to late pregnancy.
We have extended the analyses as suggested, using changes in psychological variables from baseline to 35-37 weeks as potential mediating factors for intervention effects at 35-37 weeks. Only two mediated effects were found: MVPA at 35-37 weeks was mediated by outcome expectancy of risk reduction of GDM in the PA group, and sugary drinks consumption was mediated by task self-efficacy for healthy eating in the HE&PA group. These results have been added to the paper: Table 3 is extended and the mediation results are described in the text.
6. Table 1: Please include p-values from ANOVA analysis to report any potential differences in characteristics between the groups which could potentially influence outcomes and psychological mediators. For example, the rate of smokers appears lower in the HE&PA group which could have a preferential influence on dietary intakes and compliance to PA goals.
We have added the p values of ANOVA analyses for differences between the four groups
7. Table 2: Again, reporting any potential differences in each construct across the groups at each time point would be helpful here. If there are any differences at baseline, this could influence the effectiveness of interventions. Any differences between groups at follow-up assessments could indicate effectiveness of a particular intervention.
We have added the p values of differences between groups, and indicated in bold significant differences with the usual care group.
8. Table 2 and 3: the range values for psychological constructs in parentheses should be explained. Either in the footnotes or by adding the term “range” before the numbers.
“range” has been added in both tables.
9. Results section 3.2: “Social support for healthy eating was negatively associated with portion size” should include “at 35-37 weeks” since the association was only significant at this time point.
Changed as suggested.
10. Results section end of page 4: the caption for Figure 2 appears to be incorrectly inserted here.
11. All figures: please improve layout so that all labels and text around the arrows is visible and legible.
Comments 10 & 11: We have changed the lay-out of the figures, including the captions, since all appeared to have shifted in comparison to the original word version. Hopefully, it is all legible and correct in the version that the reviewers receive.
12. Discussion: First sentence does not make sense. It seems you mean to say the following “we found that all three interventions in the DALI study were associated with an increase in outcome expectancy… compared to the control group”.
Thanks for the suggestion for improving the sentence. We have changed it accordingly.
13. Discussion: Improved self-efficacy is the dominant finding in this study in relation to mediating effects on healthy behaviors. The authors should make suggestions on how future studies that focus on lifestyle behavior change in pregnancy, or clinical practice with overweight/obese pregnant women, can incorporate efforts to increase women’s self-efficacy for more effective health outcomes. Also please reference this relevant paper in relation to how intervention studies with pregnant women can be designed to incorporate effective behavior change modalities: Ainscough et al., Public Health Nutr. 2017 Oct;20(14):2642-2648. doi: 10.1017/S136898001700129X.
Thank you for suggesting the inclusion of this paper. We have added suggestions on how to increase self-efficacy in future studies/interventions to the discussion and added the paper to the reference list.
14. Conclusion: in the first sentence, please re-write as “…of which the latter was related to improvements in some dietary and physical activity behaviors.”
Changed as suggested.

Reviewer 2 Report
This manuscript reports secondary analysis in the DALI Lifestyle Study. This single-blinded randomized study, with 4 intervention arms, examines effects of motivational interviewing on GDM risk reduction in obese women. This paper is relevant to outcomes of the increasing proportion of obese pregnancies associated with a sedentary lifestyle and unhealthy diet. From a psychology point of view it is an engaging paper as regards socio-demographic determinants of lifestyle changes in pregnancy, however I have some comments:
As this is largely a clinical journal it would help to show the trajectories of gestational weight gain, blood glucose levels and pregnancy outcomes as relevant outcomes. As about a third to half of reproductive age women begin pregnancy obese, and advise is inadequately provided by doctors and antenatal clinic nurses, a simple message for the mother beyond improved health behaviour should be considered. That would improve its relevance and translation to clinical practice.
Why isn't the vitamin D level shown as a potential confounder.
Can the authors speculate on why the physical activity group consumed more sugar and larger portions?
Table 1 and 2 - please show P value or CIs for comparisons between the 4 groups.
Interpretation of CIs is not uniformly accurate. Most are not significant.
Author Response
This manuscript reports secondary analysis in the DALI Lifestyle Study. This single-blinded randomized study, with 4 intervention arms, examines effects of motivational interviewing on GDM risk reduction in obese women. This paper is relevant to outcomes of the increasing proportion of obese pregnancies associated with a sedentary lifestyle and unhealthy diet. From a psychology point of view it is an engaging paper as regards socio-demographic determinants of lifestyle changes in pregnancy, however I have some comments:
As this is largely a clinical journal it would help to show the trajectories of gestational weight gain, blood glucose levels and pregnancy outcomes as relevant outcomes. As about a third to half of reproductive age women begin pregnancy obese, and advise is inadequately provided by doctors and antenatal clinic nurses, a simple message for the mother beyond improved health behaviour should be considered. That would improve its relevance and translation to clinical practice.
Although we fully agree with the reviewer that glucose and weight trajectories are relevant and interesting, we did not include those in this paper. Those data have been published previously (Simmons et al. 2017), and are not directly related to the main aims of this particular paper.
Why isn't the vitamin D level shown as a potential confounder.
In this paper, we report only on the DALI Lifestyle trial, in which vitamin D supplementation was not an intervention arm. We do not think that vitamin D levels in maternal serum would confound the associations between intervention and changes in psychological variables, or lifestyle outcomes, and have therefore not included it in the analyses.
Can the authors speculate on why the physical activity group consumed more sugar and larger portions?
This an interesting aspect, that we cannot really explain based on data that we have. It might be that women who are more active feel that they need to drink more (afterwards) and reward themselves with something sweetened. However, as said, we cannot base this on literature or data, and we therefore did not include it in the discussion of the paper.
Table 1 and 2 - please show P value or CIs for comparisons between the 4 groups.
We have added the p values to these tables and made differences with the usual care group in bold in Table 2.
Interpretation of CIs is not uniformly accurate. Most are not significant.
We have checked the interpretation of the CIs carefully. The reviewer is correct that many associations shown are not significant, especially the direct paths in the mediation analyses. This means that (most of) the effect on the lifestyle outcome is through the mediating variable(s).

Reviewer 3 Report
Van Poppel et al detail the analysis of how psychosocial factors may mediate lifestyle behavior changes in the DALI lifestyle RCT. They found some moderate effects of self-efficacy, and some unexpected relationships with outcome expectancy. The manuscript is well-written and results are clearly presented. Some minor changes are necessary for clarification and presentation.
Table 2 data is not discussed in the results section, but there appears to be differences between groups in some of the mediators and outcomes. If these apparent differences are not statistically significant, this should be stated clearly in the text. If there are differences, they should be highlighted in the table.
Table 3 data does include statistically significant relatioships between intervention and psychosocial variables, however it is difficult to visualize these effects in the table. Please bold the significant relationships as was done in the supplementary tables.
In Figure 1, the relationships between intervention condition and task self-efficacy for physical activity at 24-28 wks (a1, a2, a3) are different between Fig 1A and 1B, but appear to be calculated between the same variables?
The Figures are formatted poorly so that several data points are obscured, cut-off or missing. This should be corrected.
Section 4.1, the second last line, the authors state "...we did not include any items related to sedentary behavior, or to the specific healthy eating goals." It is unclear whether they are referring to psychosocial factors that specifically relate to those goals, or to lifestyle outcomes (which did include those measures: PPAQ, FFQ). Please clarify.
Author Response
Van Poppel et al detail the analysis of how psychosocial factors may mediate lifestyle behavior changes in the DALI lifestyle RCT. They found some moderate effects of self-efficacy, and some unexpected relationships with outcome expectancy. The manuscript is well-written and results are clearly presented. Some minor changes are necessary for clarification and presentation.
Table 2 data is not discussed in the results section, but there appears to be differences between groups in some of the mediators and outcomes. If these apparent differences are not statistically significant, this should be stated clearly in the text. If there are differences, they should be highlighted in the table.
We have added p values in the table and some text around the table in the results section.
Table 3 data does include statistically significant relatioships between intervention and psychosocial variables, however it is difficult to visualize these effects in the table. Please bold the significant relationships as was done in the supplementary tables.
We have added the p values and made significant differences bold.
In Figure 1, the relationships between intervention condition and task self-efficacy for physical activity at 24-28 wks (a1, a2, a3) are different between Fig 1A and 1B, but appear to be calculated between the same variables?
The numbers are slightly different in figure 1A and 1B, because of different number of participants in the analysis: 382 for MVPA at 214-28 weeks and 332 In the analysis of MVPA at 35-37 weeks.
The Figures are formatted poorly so that several data points are obscured, cut-off or missing. This should be corrected.
We have changed the lay-out of the figures, including the captions, since all appeared to have shifted in comparison to the original word version. Hopefully, it is all legible and correct in the version that the reviewers receive.
Section 4.1, the second last line, the authors state "...we did not include any items related to sedentary behavior, or to the specific healthy eating goals." It is unclear whether they are referring to psychosocial factors that specifically relate to those goals, or to lifestyle outcomes (which did include those measures: PPAQ, FFQ). Please clarify.
We have clarified that we meant items related to psychosocial factors related to these lifestyle goals.

Reviewer 4 Report
Thank you for the opportunity to review your paper. Your research is important and of high interest. However, I believe the paper could be improved considering the following points:
1. What kind of physical activity and healthy eating messages were discussed with patients in the intervention groups? Please attached them to supplementary materials.
2. Why did you advise to keep gestational weight gain below 5 kg when according to the US Institute of Medicine range of total weight gain for obese patients is 5.0-9.1?
Institute of Medicine (US) and National Research Council (US) Committee to Reexamine IOM Pregnancy Weight Guidelines; Rasmussen, K.M.; Yaktine, A.L. Weight Gain during Pregnancy: Reexamining the Guidelines; National Academies Press, Washington (DC), WA, USA, 2009.
3. In table 2 we don’t know if there are some significant differences between groups, the significance (p) results should be added.
4. Figures are out of focus, difficult to read - please correct.
Author Response
Thank you for the opportunity to review your paper. Your research is important and of high interest.
However, I believe the paper could be improved considering the following points:
1. What kind of physical activity and healthy eating messages were discussed with patients in the intervention groups? Please attached them to supplementary materials.
We have added a table with this information to the supplementary materials (Appendix 1).
2. Why did you advise to keep gestational weight gain below 5 kg when according to the US Institute of Medicine range of total weight gain for obese patients is 5.0-9.1?
Institute of Medicine (US) and National Research Council (US) Committee to Reexamine IOM Pregnancy Weight Guidelines; Rasmussen, K.M.; Yaktine, A.L. Weight Gain during Pregnancy: Reexamining the Guidelines; National Academies Press, Washington (DC), WA, USA, 2009.
The US Institute of Medicine recommended range of total weight gain for obese patients of 5.0-9.1 was a source of much discussion within our group. Danish data (1) suggested that a gestational weight gain target of <5kg was associated with less adverse outcomes with no increase in eg SGA and this was supported by further Danish data among women with type 2 diabetes (2). We were concerned about possible risks from gestational weight loss (3) and hence agreed on a target of 5 kg without gestational weight loss rather than 9kg.
(1) Jensen DM, Ovesen P, Beck-Nielsen H, Mølsted-Pedersen L, Sørensen B, Vinter C, Damm P. Gestational Weight Gain and Pregnancy Outcomes in 481 Obese Glucose-Tolerant Women. Diabetes Care 2005; 28(9):2118-2122. DOI: 10.2337/diacare.28.9.2118
(2) Ásbjörnsdóttir B, Rasmussen SS, Kelstrup L, Damm P, Mathiesen EM. Impact of Restricted Maternal Weight Gain on Fetal Growth and Perinatal Morbidity in Obese Women With Type 2 Diabetes. Diabetes Care 2013;36(5):1102-1106. DOI: 10.2337/dc12-1232
(3) Catalano PM, Mele L, Landon MB, Ramin SM, et al. Inadequate weight gain in overweight and obese pregnant women: what is the effect on fetal growth? Am J Obstet Gynecol 2014; 211(2): 137.e1-7. DOI: 10.1016/j.ajog.2014.02.004
3. In table 2 we don’t know if there are some significant differences between groups, the significance (p) results should be added.
We have added the p values to this table.
4. Figures are out of focus, difficult to read - please correct.
We have changed the lay-out of the figures, including the captions, since all appeared to have shifted in comparison to the original word version. Hopefully, it is all legible and correct in the version that the reviewers receive.

Round 2
Reviewer 1 Report
The authors have addressed all comments thoroughly. The manuscript is now improved and meets requirements for publication.
Author Response
Thank you for the positive feedback.
Reviewer 2 Report
Thank you for considering my comments. This is an improved version of the manuscript. Can you include a comment on why 5kg weight gain limit was decided upon? Also, include Fisher's exact test in the Chi square stats plan.
Author Response
Reviewer 2
Thank you for considering my comments. This is an improved version of the manuscript. Can you include a comment on why 5kg weight gain limit was decided upon? Also, include Fisher's exact test in the Chi square stats plan.
We have included the following comment in the text on the 5 kg weight gain limit:
The 5 kg weight gain target - without gestational weight loss - rather than the 9 kg recommended by the Institute of Medicine [19] was chosen since it was associated with less adverse outcomes in previous studies [20,21].
Furthermore, we have added the Fisher’s exact test in the description of the statistical analyses.